# Microstructural and Strength Changes in Trabecular Bone in Elderly Patients with Type 2 Diabetes Mellitus

**DOI:** 10.3390/diagnostics11030577

**Published:** 2021-03-23

**Authors:** Mercè Giner, Cristina Miranda, María Angeles Vázquez-Gámez, Patricia Altea-Manzano, María-José Miranda, Antonio Casado-Díaz, Ramón Pérez-Cano, María-José Montoya-García

**Affiliations:** 1Departamento de Citología e Histología Normal y Patológica, Universidad de Sevilla, Avda. Dr. Fedriani s/n, 41009 Sevilla, Spain; 2Bone Metabolism Unit, Department of Internal Medicine, HUV Macarena, Avda Sánchez Pizjuán s/n, 41009 Seville, Spain; crismirandadiaz@yahoo.es (C.M.); mavazquez@us.es (M.A.V.-G.); m.j.mir@telefonica.net (M.-J.M.); rpcano@us.es (R.P.-C.); pmontoya@us.es (M.-J.M.-G.); 3Medicine Department, University of Seville, Avda. Dr. Fedriani s/n, 41009 Sevilla, Spain; patriciaaltea@gmail.com; 4IMIBIC (Instituto Maimónides de Investigación Biomédica de Córdoba), Hospital Universitario Reina Sofía, Universidad de Córdoba & RETICEF, CIBERFES (ISCIII), 14004 Córdoba, Spain; bb1cadia@uco.es

**Keywords:** trauma, fracture, hip, diagnostic, imaging, osteoporosis, type 2 diabetes mellitus

## Abstract

Type 2 diabetes mellitus (T2DM) is one of the most common chronic diseases worldwide and it is associated with an increased risk of osteoporosis and fragility fractures. Our aim is to analyze the effect of T2DM on bone quality. This is a case-control study. The studied population consisted of 140 patients: 54 subjects with hip fracture (OP) without T2DM, 36 patients with hip fracture and T2DM (OP-T2DM), 28 patients with osteoarthritis (OA) without T2DM, and 22 patients with OA and T2DM (OA-T2DM). Bone markers, bone mineral density, FRAX score, microstructural, and bone material strength from femoral heads were assessed. The group with hip fracture presented lower BMD values than OA (*p* < 0.05). The OP, OP-T2DM, and OA-T2DM groups showed a decrease in bone volume fraction (BV/TV), in trabecular number (Tb.N), and in trabecular thickness (Tb.Th), while an increase was presented in the structural model index (SMI) and trabecular bone pattern factor (Tb.Pf), The groups OP, OP-T2DM, and OA-T2DM also presented lower values than those in group OA regarding the biomechanical parameters in the form of Young’s modulus or elastic modulus, toughness, ultimate stress, ultimate load, extrinsic stiffness, and work to failure (*p* < 0.05). Our results show the negative effect of type 2 diabetes mellitus on trabecular bone structure and mechanical properties.

## 1. Introduction

Type 2 diabetes mellitus (T2DM) is one of the most common chronic diseases worldwide. T2DM patients exhibit an increased risk of suffering further complications of the disease, which are mainly due to complex and interconnected mechanisms, such as hyperglycaemia, insulin-resistance, low-grade inflammation, and accelerated atherogenesis [1]. In addition, these chronic complications adversely affect multiple organ systems including that of bones, which were widely associated with an increased risk of osteoporosis and fragility fractures [2,3].

Osteoporosis (OP) is a metabolic bone disease that is characterized by low bone mineral density (BMD) and microarchitectural deterioration in the bone structure, with a higher risk of fragility fractures [4]. Osteoarthritis (OA) is a metabolically active and dynamic process that involves all joint tissues. OA can lead to mechanical failure, pain, and surgical joint replacement with a prosthesis [5]. Clinical and epidemiological studies suggested that OP and OA may coexist in the same patient. BMD values are usually normal or elevated in OA patients at any age, in contrast to OP [6].

T2DM is also very frequent in elderly people, and it exhibits high morbidity and mortality. Furthermore, various lifestyle factors contribute towards the increased incidence of T2DM, OA, and OP [7,8]. On the other hand, patients with T2DM are at significant risk regarding fragility fractures at skeletal sites, such as the hip, spine, and forearm, although these patients often have normal or increased BMD [9,10]. A published meta-analysis showed that T2DM patients exhibited a relative risk of 1.7 (95% CI: 1.3 to 2.2) for hip fracture, and surprisingly, BMD was generally higher in patients with T2DM [11]. The low cortical bone strength and poor bone quality, due to the glycation of bone proteins, may also underlie increased fracture risk in diabetes [12].

The greater risk of an osteoporotic fracture despite normal or high BMD [13] and makes it necessary to identify the factors that influence this risk and elucidate diabetes-induced alterations in trabecular bone microarchitecture and bone turnover markers.

The goal of this study was to analyze trabecular microstructural and mechanical properties from femoral heads using micro-CT, BMD and bone turnover markers (BTM) in patients with T2DM with or without recent fragility fractures in order to examine skeletal outcomes related to this disease and to establish a relative risk assessment method in the clinical setting. In this regard, we hypothesized that diabetic patients have higher BMD values than expected, and that T- score and FRAX can predict fracture but underestimate risk, and therefore the estimate of the skeletal properties of these patients should be adjusted.

## 2. Materials and Methods

### 2.1. Study Design and Subjects

The study was designed as a case-control study. Subjects were included in a consecutive manner (October 2019 to July 2020). The sample size was calculated with the Granmo sample size and power calculator (v.7.12, IMIM, Barcelona, Spain) (https://www.imim.es/ofertadeserveis/software-public/granmo/), in order to detect a significant standardized mean difference of 0.5 (one size average effect) in bone volume fraction (BV/TV) with a type I error rate of 5% (alpha = 0.05) and a 90% power (1-beta = 0.90). Therefore, 7 subjects were required in each group.

The population studied consisted of 140 patients (aged 65–93), of which the case study group was made up of 90 patients with hip osteoporotic fracture (OP group) undergoing a prosthetic hip replacement, and the control group featured the 50 patients with hip osteoarthritis (OA group), but without ever having suffered from an osteoporotic fracture that would need undergoing total hip arthroplasty. Both groups were subdivided according to the criteria of whether the patient had T2DM. The case-study group was then subdivided further into OP without T2DM and OP-T2DM as was the control group, which was divided into OA without T2DM and OA-T2DM (Figure 1). Two patients were excluded: 2 in the OP group due to their lack of a normal kidney function.

The inclusion criterion for the OP patients was the possession of a current frailty hip fracture (a fall from less than the patient’s height without an acceleration mechanism). For inclusion, the OA patients could not have been previously diagnosed with osteoporosis nor could they have a history of frailty fracture from the age of 50, nor could they have congenital or acquired dysplasia or avascular necrosis.

The medical history of all patients was checked and the diagnosis of T2DM for more than 5 years was confirmed for the diabetic patients and discarded for the rest of them. All of the T2DM patients have been treated with metformin.

The exclusion criteria for all the groups included malignant diseases, hyperthyroidism, hyperparathyroidism, multiple myeloma, rheumatoid arthritis, osteomalacia, secondary OP due to corticosteroids or those who were treated with osteoporosis drugs. In addition, patients with congenital or acquired dysplasia or avascular necrosis were excluded from the OA and OA-T2DM groups. Both groups of patients had normal kidney function.

Standardized interviews were employed to obtain the following information: age (years), weight (kg), height (cm), body mass index (BMI) (kg/m^2^), use of calcium and vitamin D supplements (none vs. any), use of medication, and family history of hip fracture.

We estimated the 10-year risk of major osteoporotic fracture (clinical spine, hip, forearm, or humerus fracture) and the 10-year risk of hip fracture for each patient by means of the FRAX^®^ tool, calibrated for Spain (www.shef.ac.uk/FRAX/index.htm, accessed on June 2020) (University of Sheffield, Sheffield, UK). The criteria of the Scientific Advisory Council of Osteoporosis in Canada was employed to classify the FRAX^®^ scores [14].

The study was approved by the Ethical Review Board of Seville (internal references 2147), and was performed in accordance with the ethical standards as laid down in the 1964 Declaration of Helsinki and its later amendments. Written informed consent was obtained from all participants. All patients included in the study agreed to donate their bone samples. Arthroplasty was performed in the Orthopaedics & Traumatology Department of the “Virgen Macarena” University Hospital (Seville, Spain). Due to the difficulty in obtaining healthy donors of hip bones, the osteoarthritic samples (OA) were considered as the reference group since the BMD values classified them as non-osteoporotic.

### 2.2. Biochemical Measurements

Fasting morning blood was drawn and stored at −80 °C. We assessed carboxy-terminal telopeptide of type I collagen (CTX), aminoterminal propeptide of type I procollagen (P1NP), parathyroid hormone (PTH), 25-hydroxyvitamin D (25(OH)D), insulin-like growth factor I (IGF-I), and glycated haemoglobin (HbA1c).

PTH, CTX, and P1NP were analyzed by immunoassay on an autoanalyzer COBAS 601 (Roche, Spain); inter-assay was CV <5.8%, <7.6%, and <4.2%, respectively.

25(OH)D was analyzed by direct competitive immunoassay on an autoanalyzer LIAISON (DiaSorin, Saluggia, Italy); inter-assay was CV <5.5%. IGF-I was determined using a chemiluminescence immunoassay (CLIA) by an autoanalyzer (IMMULITE 2000, Siemens, Erlangen, Germany); inter-assay CV was 6.9%. HbA1c was measured using an autoanalyzer (ADVIA 2400, Siemens, Erlangen, Germany; inter-assay CVs was 1%). In all cases, the intra-assay CV was <5%.

### 2.3. Bone Mineral Density

BMD of the total hip and femoral neck was measured through Dual-energy X-ray absorptiometry on the Hologic Discovery W densitometer using the APEX 3.1.1 software (Hologic, Inc., Waltham, MA, USA). In vivo CV was 2.4% (femoral neck) and 1.1% (total hip).

### 2.4. Microstructural and Bone Material Strength

The bone samples were donated from parts of the patients who participate in this study and may reflect the characteristics of the serum index. The case-study group was OP without T2DM (6 patients) and OP-T2DM (8 patients), OA without T2DM (7 patients) and OA-T2DM (7 patients).

Femoral heads were stored frozen in phosphate-buffered saline (PBS, Lonza, Basel, Switzerland) at −20 °C until processing. A cylinder of trabecular bone was extracted from each femoral head and processed as previously optimized and described by our laboratory [15]. The bone cores were analyzed by micro-Ct without further preparation (Skyscan 1172, Bruker micro-CT NV, Kontich, Belgium). The following histomorphometric parameters were measured: bone volume fraction (BV/TV; %), bone surface density (BS/TV; 1/mm), trabecular thickness (Tb.Th; mm), trabecular separation (Tb.Sp; mm), trabecular number (Tb.N; 1/mm), structural model index (SMI), and trabecular bone pattern factor (Tb.Pf; 1/mm).

In order to visualize material failure on a microstructural level, compression tests of the bone cores were performed using a micro-mechanical testing device (Material Testing Stage, Bruker micro-CT NV, Kontich, Belgium). The following mechanical parameters were measured: ultimate load (Fult; N), extrinsic stiffness (S; N/mm); work to failure (U; mJ); and intrinsic or material mechanical properties including: ultimate stress (σult; MPa); Young’s modulus (E; MPa); and toughness (u; MPa).

### 2.5. Statistical Analysis

Continuous variables are presented as the mean and standard deviation (SD). In order to compare continuous variables with a normal distribution, ANOVA was utilized with more than two samples, and the Student’s *t*-test with two samples; if the distribution was not normal, then the Kruskal–Wallis test or the Mann–Whitney *U* test was used. Normality testing was performed using a combination of the Kolmogorov–Smirnov tests. An adjustment for age, the analysis of covariance (ANCOVA), was applied in the OA group when needed (to compare microstructural and bone material strength parameters). Correlations between variables were examined using the Spearman correlation, which is appropriate for smaller sample sizes with robust potential outliers. All hypotheses were two-tailed, and a *p* value < 0.05 was considered statistically significant. Statistical package SPSS 20.0 for Windows (IBM Corp., Armonk, NY, USA).

## 3. Results

In order to ascertain whether type 2 diabetes mellitus has a related effect on bone fragility and risk of fracture, we compared the microstructural and biomechanical parameters of osteoporotic patients (OP) either with type 2 diabetes mellitus (OP-T2DM group) or without T2DM (OP group), both with recent fragility fractures, and we also compared the OA and OA-T2DM group patients. Due to the difficulty in using bone samples from heathy donors, osteoarthritis patients (OA group) were used as the non-osteoporotic control, in the same way as in other studies were they were employed [15]. In Table 1 and Table 2, we observed the anthropometric characteristics, BMD, and biochemistry of all the groups. No significant difference between the values by gender could be found, and hence we performed the statistical analysis with the total means of the group.

Since OA patients were younger (73.4 ± 6.2 years) than the other groups, an adjustment for age was applied to the successive statistical analyses. No significant differences were found in weight, height, or BMI between the study groups (Table 1 and Table 2). Densitometric parameters were higher in the OA and OA-T2DM groups. Significant differences in BMD at the femoral neck and total hip were observed between OA and OP (*p* = 0.000; *p* = 0.000), OA vs. OP-T2DM (*p* = 0.003; *p* = 0.001), and between OA-T2DM vs. OP-T2DM (*p* = 0.000; *p* = 0.000). This indicated that fracture patients have worse BMD values, both for total hip and for femoral neck, and that OA patients, regardless of whether they have T2DM, present BMD in normal ranges. The FRAX^®^ 10-year risk of major or hip fracture was higher in OP patients than in OA patients (*p* = 0.026) and with T2DM (*p* = 0.000). The FRAX^®^ 10-year risk of major and hip fracture between the OP and OP-T2DM groups and the OA and OA-T2DM patients were statistically not significant.

The serum levels of the bone turnover markers were also analyzed in all groups. β-CrossLaps was lower in the OA-T2DM patients than the other groups (*p* < 0.013). On the other hand, P1NP values were lower in the OP-T2DM group compared to the OP group, reaching almost the limit of statistical significance (*p* = 0.055). In addition, we verified that the HbA1C values of the 91 patients evaluated were positively and significantly correlated with BMD of the femoral neck (*r* = 0.313) and total hip (*r* = 0.296), (*p* < 0.05), and negatively with β-CrosLapps (*r* = −0.316; *p* = 0.025). Vitamin D levels were below 20 ng/mL in all groups of patients, without significant differences between groups (Table 1 and Table 2).

Three-dimensional reconstruction and micro-CT images are shown in Figure 2. It can be observed how the samples of the OA group show a higher BMD than that in the other groups and how the effect of T2DM produces a significant deterioration at the macroscopic level in the trabecular bone structure.

Microstructural indices showed differences in the cancellous bone microarchitecture between groups. Most of the studied parameters in osteoporotic subjects, with and without T2DM, and OA-T2DM subjects showed significant differences compared to the control group (OA group) (Figure 3). The OP, OP-T2DM, and OA-T2DM bone samples had smaller BV/TV compared to the OA bone biopsies (−53%, *p* = 0.001; −36%, *p* = 0.01; −53%, *p* = 0.001). At the trabecular level, the OP group, OP-T2DM, and OA-T2DM showed a lower number of trabeculae, TbN (−33%, *p* = 0.0013; −32%, *p* = 0.233; −14% *p* = 0.016), and less width of the trabeculae Tb.Th (−37%, *p* = 0.001; −35%, *p* = 0.001; −27%, *p* = 0.000). There were no significant differences with the separation of the trabeculae, although there does tend to be fewer in all of them compared to the OA group, Tb.Sp (−10, −15, and −5%). Furthermore, these three groups also had higher Tb.Pf (in all groups almost 6 times more *p* < 0.02). Between the OP group vs. OP-T2DM, we found significant differences in the SMI parameter (OP: 1.7 ± 0.16; OP-T2DM 1.1 ± 0.16, *p* = 0.015), the other microstructural indices presented no significant differences. These results showed that the OA-T2DM patients had similar microstructural characteristics to those of the OP and OP-T2DM patients regarding trabecular bone characteristics, which indicated that the diabetic disease maintains BMD, but the microarchitecture is of poor quality and more fragile.

In order to test whether the strength of the trabecular bone was lower in T2DM patients (OA-T2DM and OP-T2DM) and in OP patients compared with OA patients, various biomechanical parameters were studied (Figure 4). The OA-T2DM, OP-T2DM, and OP samples showed lower stiffness due to the structural features and material properties of bone: Young’s modulus (−59, −38, and −51%, respectively; *p* < 0.022) and ultimate stress (−61, −46, and −68%; *p* < 0.001). Toughness (−34, −51, and −80%, respectively; *p* = 0.021), work to failure (−61, −54 and −80%, respectively; *p* = 0.005), extrinsic stiffness (−58, −38, and −49%, respectively), and ultimate load (−65, −49, and −68%, respectively; *p* < 0.001) were lower in T2DM groups and the OP group compared with the OA group. The OA-T2DM patients exhibited similar biomechanical parameters to patients with osteoporotic hip fracture (OP) in the trabecular bone, while the OP-T2DM group showed their most affected values in the biomechanical parameters. The statistical power for significant differences (*p* < 0.05) for each of the variables studied was calculated, and a higher statistical power of 88% (beta error < 12%) was consistently found.

Finally, no statistically significant correlations were encountered between the duration of T2DM and serum levels of HbA1c with microstructural values.

## 4. Discussion

It is widely accepted by the scientific community that T2DM impairs bone metabolism [8,16], and the risk of fragility fractures is increased in these patients [9,10]. The mechanisms by which the risk of fracture is increased, and this impairment occurs remain unclear. In addition, BMD measurements and the FRAX^®^ tool cannot predict these risks. There is therefore a need to establish a relative risk assessment method in the clinical setting in patients with T2DM for the prediction of possible impacts on bone fracture related to the disease. The purpose of our analysis was to study the impact in the microstructural and bone mechanics in T2DM patients with or without recent fragility fractures and the relationship of these fractures with type 2 diabetes mellitus.

To verify the influence of T2DM on quantity and quality of trabecular bone tissue from 28 patients, we assessed BMD, microarchitecture, and biomechanical properties of femoral heads from four patient groups: OA, OP, OA-T2DM, and OP-T2DM. As expected, osteoporotic patients with hip fractures had lower BMD values than osteoarthritic patients, but no differences between osteoporotic patients with and without T2DM were found, neither were they found between the OA and OA-T2DM groups. Previous data demonstrated that T2DM patients have normal or increased BMD values [17,18,19], even when this variable was normalized by the BMI [20]. Although it can be prevalent in juveniles, T2DM is very common in the elderly, and it frequently coexists with age-related bone loss [17]. Therefore, the establishment of risk factors for fragility fractures during ageing should be identified since these factors can contribute to the risk of fracture in older diabetic patients. In our case, OP-T2DM and OP patients showed BMD values lower than −2.5T (T-score score < −2.5) at all hip sites, which was consistent with the stablished World Health Organization definitions. We found no differences in BMD between osteoarthritic groups with and without T2DM, and the values of BMD were considered normal or healthy according to the T-scores in these groups. As our BMD values showed, clinical and epidemiological studies suggested an inverse relationship between many parameters studied in OP and OA patients [6,21,22,23,24]. The increased BMD may minimize the expected negative effects on bone metabolism caused by diabetes.

The most relevant data obtained in the present study was based on the decrease in the values observed of the bone microarchitecture and biomechanical properties that were tested in the trabecular hip bone of patients with both osteoarthritis and T2DM compared with non-diabetic osteoarthritic patients. Both groups of osteoarthritic patients were similar in terms of age, weight, lifestyle, and evolutionary stage of the degenerative disease. However, bone strength in the T2DM group was substantially damaged, and remained in a similar range in patients with osteoporotic hip fracture. These results show the negative effect of T2DM on trabecular bone structure and mechanical properties. Recent data of postmenopausal women with T2DM demonstrated lower BMD values and trabecular bone microarchitecture compared to women without T2DM [18], which was consistent with our results. Conversely, improved properties of trabecular bone were noted in postmenopausal women with T2DM compared with controls; however, compromised cortical bone microarchitecture (e.g., increased cortical porosity) [25] was observed. Cortical bone characteristics were not evaluated in this study, but damage in the bone microstructure and its mechanics were demonstrated: these are important elements in trabecular bone quality in these subjects with T2DM [26,27]. We also showed a significant deterioration in these parameters in patients with hip fractures. However, a lower quality of trabecular bone was not observed in OP-T2DM patients compared to non-diabetics. Patsch et al. showed similar results in younger people with diabetes using HR-pQCT of the ultradistal and distal radius and tibia [28]. This data indicated that diabetic disease is a key factor directly involved in the deterioration of bone quality, which is likely responsible for the increased risk of fragility fractures.

We observed positive correlation between HbA1C and BMD of the femoral neck, HbA1C with BMD of the total hip and negatively with β-Croslapp (*r* = −0.316) (*p* = 0.025). These results were in line with a decrease in bone remodeling in diabetic patients and its contribution to the alteration of bone quality, more than in the quantity of bone. OP subjects exhibited bone remodeling of a more active nature, primarily due to bone resorption, as evidenced by the significantly higher levels of β-CrossLaps, which was consistent with a previous study [15]. However, a trend towards reduced bone remodeling activity was observed in T2DM patients, which was demonstrated by the lower levels of formation and resorption markers than in the respective controls [18,19,29,30]. Certain authors reported defects in bone formation that were produced by a decrease in osteoblast differentiation by an increase in apoptosis in these cells [31]. These changes may lead to an imbalance between bone resorption and bone formation [30,32].

Considerable evidence suggested that specific factors, such as poor glycaemic control and T2DM duration (e.g., a glycated haemoglobin level ≥7.5%) [33] exacerbate risk factors in T2DM patients, although this relationship has not been established unequivocally [34]. In the present study, both groups of diabetic patients were fairly well controlled. The average glycated haemoglobin level was lower than the previously mentioned average [33], and it was not associated with mechanical or microstructural parameters. We found no correlation between these values and the duration of diabetic disease, which remained very similar in the OP-T2DM and OA-T2DM groups, and was consistent with a previous study [18].

The FRAX tool revealed that the OP and OP-T2DM subjects showed moderate to high risk. However, OA-T2DM patients of similar age showed low risk probability for both types of fracture, but mechanical and microstructure indicated the opposite risk. These results suggested that effective intervention thresholds for fracture prevention in patients with T2DM might be different than those that are effective for non-diabetic patients, as discussed recently [28,35].

Despite the potential of the current data obtained from human hip bone samples, our study had several limitations. Our sample size was relatively small although with sufficient power to obtain significant differences between the differences found. Therefore, our results should eventually be confirmed. The lack of cortical bone microarchitecture and histomorphometric indices of the samples is an additional concern. Nevertheless, this study includes BMD, BTM, trabecular bone microarchitecture, and mechanical strength measures in T2DM patients with and without having recently suffered fragility hip fractures, thus allowing for the establishment of a relationship between osteoporotic fracture and diabetes.

## 5. Conclusions

Our findings were the first demonstration of compromised trabecular bone microarchitecture and material properties using measurements of human bone biopsies from the hip with and without fragility fractures in T2DM subjects. In conclusion, we showed the potential detrimental effects of diabetic disease on bone quality. These findings highlighted the importance of evaluating diabetic patients using bone markers and bone quality parameters. Diabetes should therefore be included as a risk factor for osteoporotic fracture in daily clinical practice.

## Figures and Tables

**Figure 1 diagnostics-11-00577-f001:**
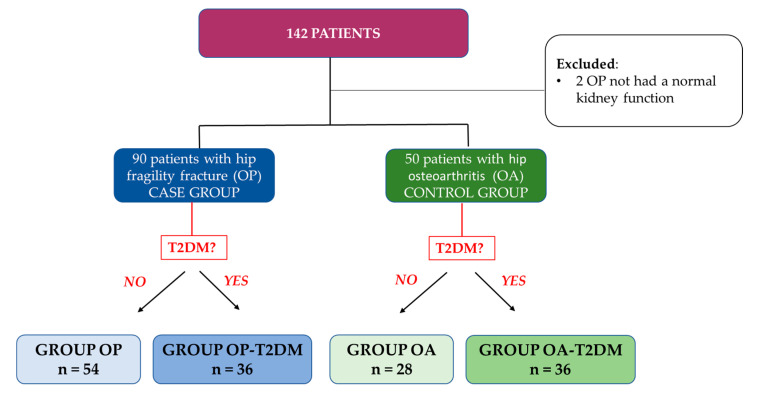
Outline of patient selection and study groups. OP, osteoporotic; OA, osteoarthritis; T2DM, type 2 diabetes mellitus.

**Figure 2 diagnostics-11-00577-f002:**
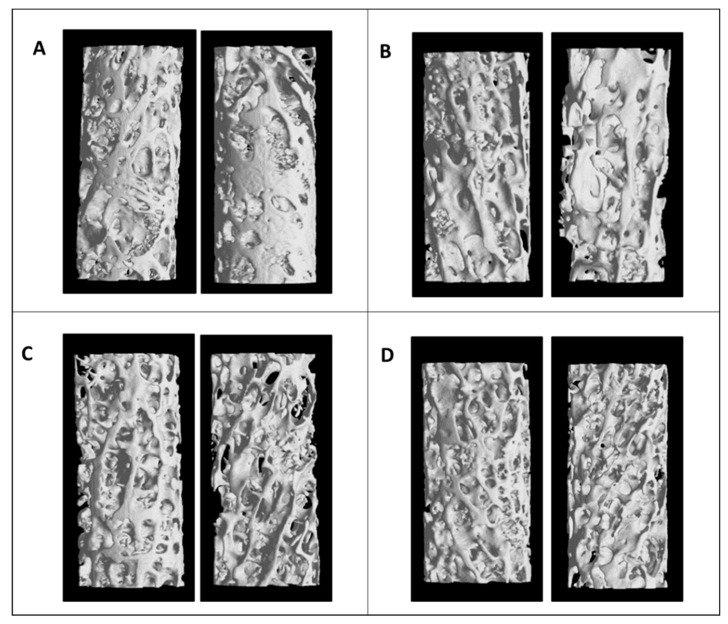
Three-dimensional reconstruction and micro-CT images of bone trabecular from femoral heads in the four groups. Two different samples from: (**A**) OA group; (**B**) OA-T2DM; (**C**) OP; (**D**) OP-T2DM.

**Figure 3 diagnostics-11-00577-f003:**
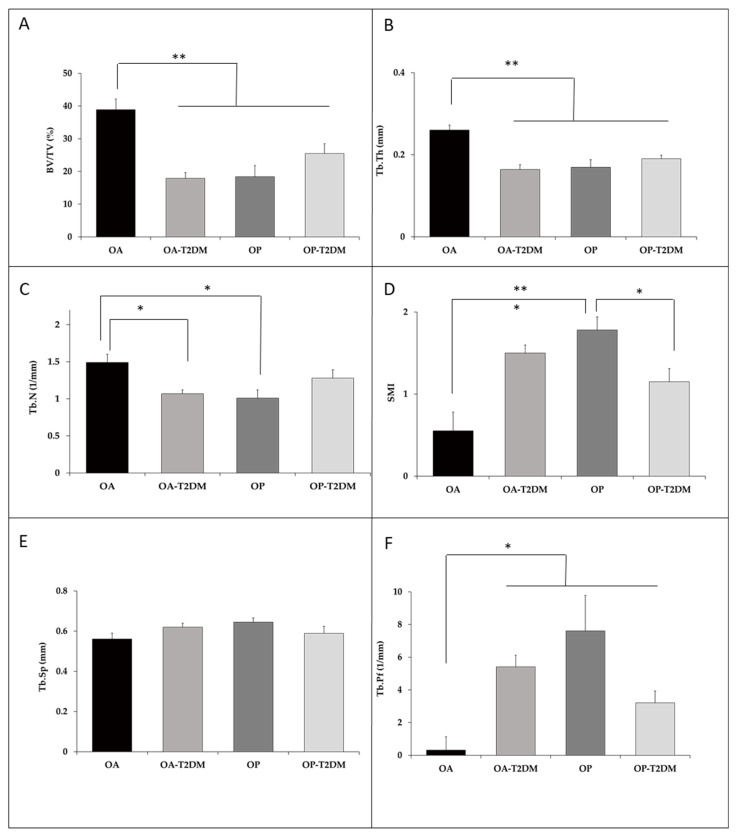
Comparisons of structural parameters between patients with OA and OP with or without T2DM. (**A**) BV/TV, bone volume fraction; (**B**) Tb.Th, trabecular thickness; (**C**) Tb.N, trabecular number; (**D**) SMI, structural model index; (**E**) Tb.Sp, trabecular separation; (**F**) Tb.Pf, trabecular bone pattern factor; Values are expressed as the means ± SEM. * *p* < 0.05, ** *p* < 0.01, statistically significant.

**Figure 4 diagnostics-11-00577-f004:**
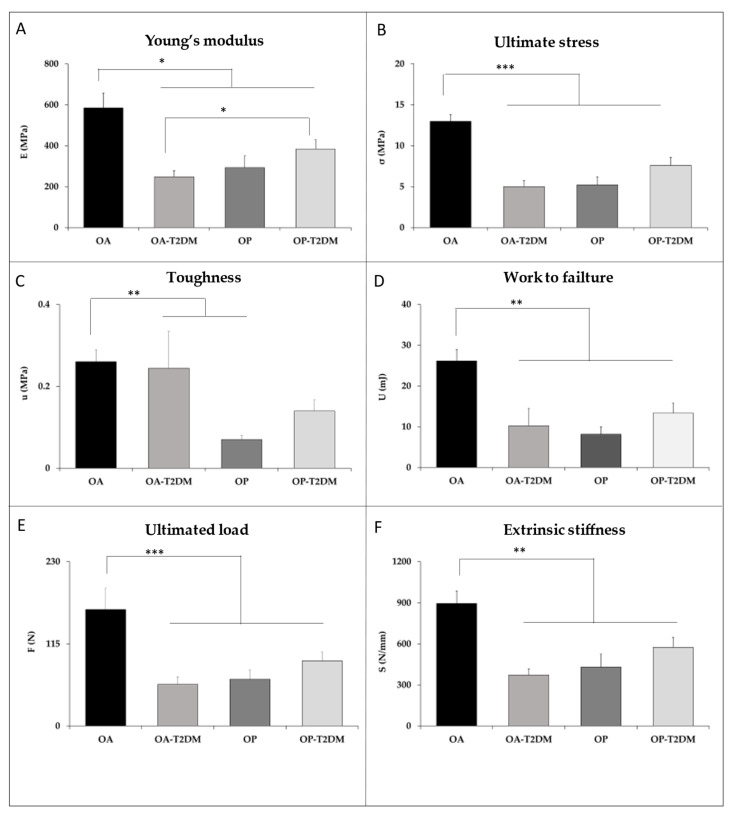
Comparisons of biomechanical parameters between patients with OA and OP.(**A**) E, Young’s modulus or elastic modulus; (**B**) σ_ult_, ultimate stress; (**C**) u, toughness; (**D**) U, work to failure; (**E**) F_ult_, ultimate load; (**F**) S, extrinsic stiffness. Values are expressed as the means ± SEM. * *p* < 0.05, ** *p* < 0.01, and *** *p* < 0.001 statistically significant.

**Table 1 diagnostics-11-00577-t001:** Patient characteristics (total and by gender) and analyzed parameters. Values are shown as the means ± standard deviation (SD). BMI, body mass index; BMD, bone mineral density; PTH, parathyroid hormone; IGF-1, insulin-like growth factor I; HbA1c, glycated haemoglobin.

	OA	OA-T2DM	*p*
Gender (Male/Female)	28 (10/18)	22 (8/14)	-
Age (years) Female Male	73.4 ± 6.2 73.4 ± 7.4 73.4 ± 5.7	74.5 ± 6.4 75.1 ± 6.7 73.5 ± 6.3	0.528
BMI (kg/m^2^) Female Male	30.7 ± 3.8 29.5 ± 3.5 32.8 ± 3.5	31.7 ± 5.1 32.3 ± 4.1 30.5 ± 7.0	0.459
Femoral neck BMD (gHA/cm^2^) Female Male	0.685 ± 0.12 0.654 ± 0.12 0.746 ± 0.09	0.756 ± 0.15 0.712 ± 0.15 0.843 ± 0.07	0.080
Hip BMD (gHA/cm^2^) Female Male	0.909 ± 0.13 0.868 ± 0.12 0.991 ± 0.09	0.944 ± 0.16 0.896 ± 0.16 1.04 ± 0.10	0.405
10-year risk of major fracture (FRAX^®^ tool)	9.3 ± 6.8	6.3 ± 5.4	0.183
10-year risk of hip fracture (FRAX^®^ tool)	3.5 ± 3.2	4.1 ± 7.1	0.772
25-hydroxyvitamin D (ng/mL)	21.8 ± 15.5	15.3 ± 6.6	0.076
PTH (pg/mL)	54.5 ± 23.4	59.5 ± 28.1	0.545
β-CrossLaps (µg/mL)	0.34 ± 0.17	0.33 ± 0.21	0.869
P1NP (ng/mL)	43.28 ± 20.3	49.2 ± 31.7	0.468
IGF-1 (ng/mL)	82.4 ± 34.4	88.8 ± 42.4	0.592
HbA1c (%)	5.5 ± 0.3	7.2 ± 1.4	0.000

**Table 2 diagnostics-11-00577-t002:** Patient characteristics (total and by gender) and analyzed parameters. Values are shown as the means ± typical deviation (SD).

	OP	OP-T2DM	*p*
Gender (Male/Female)	54 (9/45)	36 (8/28)	-
Age (years) Female Male	78.2 ± 6.7 78.3 ± 647 77.6 ± 8.4	79.1 ± 6.2 79.3 ± 5.7 78.2 ± 8.2	0.538
BMI (kg/m^2^) Female Male	28.3 ± 4.3 28.8 ± 4.4 26.6 ± 2.6	29.6 ± 5.2 29.4 ± 6.0 30.1 ± 2.9	0.256
Femoral neck BMD (gHA/cm^2^) Female Male	0.569 ± 0.09 0.566 ± 0.08 0.588 ± 0.11	0.596 ± 0.09 0.572 ± 0.09 0.681 ± 0.06	0.183
Hip BMD (gHA/cm^2^) Female Male	0.758 ± 0.10 0.747 ± 0.09 0.817 ± 0.10	0.794 ± 0.11 0.777 ± 0.09 0.884 ± 0.10	0.122
10-year risk of major fracture (FRAX^®^ tool)	13.7 ± 8.8	12.4 ± 8.7	0.501
10-year risk of hip fracture (FRAX^®^ tool)	6.3 ± 6.3	5.8 ± 6.4	0.702
PTH (pg/mL)	68.1 ± 41.1	71.3 ± 41.5	0.746
β-CrossLaps (µg/mL)	0.66 ± 0.37	0.51 ± 0.26	0.060
P1NP (ng/mL)	63.8 ± 41.7	46.3 ± 25.7	0.055
IGF-1 (ng/mL)	78.4 ± 34.7	65.9 ± 30.9	0.163
HbA1c (%)	5.4 ± 0.3	6.7 ± 1.4	0.000

## Data Availability

The data presented in this study are available on request from the corresponding author.

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
