# Peer review of "Microstructural and Strength Changes in Trabecular Bone in Elderly Patients with Type 2 Diabetes Mellitus"

_diagnostics, 2021, doi:10.3390/diagnostics11030577_

Round 1

Reviewer 1 Report

The author still did not make much progress in improving the sample number. It is understandable that the hip bone sample is extremely difficult to acquire in patients. But for data not based on the bone sample, like Table 1 and Table 2, the small sample number and mixture of male and female is not appropriate. The author stated they did not observe difference between males and females. This is just because of the low sample number. How can you get difference with only 2 male people? For example, in Table 1, the only different parameter is HbA1c, but can you make a conclusion to say HbA1c is the main reason for lower bone mass in OA-T2DM? Apparently not, because the variation in the measurement of 25-hydroxyvitamin D, PTH, β-CrossLaps, P1NP and IGF-1 are too big. It is OK to make further study and conclusion by using small group of bone samples, but for the Tables, these number of patient sample is not acceptable.

Reviewer 2 Report

Thanks for addressing all reviewer's comments. I felt majority of my comments were answered except for the following item.

R.C. Once again, conclusion regarding any null findings in this study cannot be supported secondary to very small sample size (high risk of type 2 error), such as the statement in the conclusion "Furthermore, our results in OA-T2DM subjects confirm that diabetes does not impair BMD."

A.R. The sample size is correct, it would undoubtedly be better to have more patients, but obtaining human bone biosias is quite difficult and there are few studies with a population larger than ours with human hips. The results are conclusive at the statistical level.

Since bone biopsy does not add any value to BMD assessment, non-statistically significant difference in BMD between OA & OA-T2DM and OP & OP-T2DM are not conclusive at the statistical level due to the following reasons.

  1. likely the smallest sample size than any other studies on BMD assessment that have been published; therefore, type 2 error are extremely high. I strongly disagree with authors' response on this subject stating that the null findings on BMD comparison are conclusive at the statistical level!
  2. At least a dozen of studies with larger sample size suggested increased BMD in T2DM (e.g. In T2DM, many studies including one meta-analysis have not found decreased BMD and some have shown paradoxically increased BMD, although it does not mean increased BMD is protective. In the Women’s Health Initiative women with T2DM had statistically significant increases in BMD at the spine and hip compared to women without diabetes throughout 9 years of follow up. ref: Valderrábano RJ, et al. Diabetes mellitus and bone health: epidemiology, etiology and implications for fracture risk stratification. Clin Diabetes Endocrinol 4, 9 (2018). ), I find difficulty buying in authors' statement that this null findings on BMD comparison with total sample size of 6-7 for each group are conclusive at the statistical level.
  3. Numerous data suggested the reduction in BMD in T1DM, the current study only included patients with T2DM, it is inappropriate for authors to state that the results in OA-T2DM confirm that diabetes (including T1DM) does not affect BMD.
  4. While examining the table 1 & 2, numeric differences in femoral neck BMD were seen in both OA and OP groups comparison with lower BMD in T2DM for both genders, although ns at this level of sample sizes, type 2 error cannot be excluded. This needs to be listed as one of limitations.

In summary, suggest to change the statement to "Furthermore, our results in OA-T2DM subjects suggests that type 2 diabetes may negatively affect bone quality, but not BMD or bone quantity. "

In line 162 it should be "due to" instead of "dur to"

Reviewer 3 Report

The paper presented a nice addition to the literature on T2DM on bone quality vs. quantity using human biopsy samples with/out Fracture in OA and OP patients

Author Response

Reviewer #3:

Comments and Suggestions for Authors:

The paper presented a nice addition to the literature on T2DM on bone quality vs. quantity using human biopsy samples with/out Fracture in OA and OP patients

A.R. We thanks the reviewer’s comment.

Round 2

Reviewer 1 Report

Mercè Giner et al. have added a greater number of patients to analyze the serum parameters in the response letter. Thus, the Table 1 and Table 2 content has been substantially improved. The new version Table 1 and Table 2 are better for the paper since they reflect the real change with bigger sample number. I agree with the author that the importance and strength of this study is the precious human bone biopsies which allows us to "in situ" the bone microstructure of each selected group. In the paper, you can say the bone sample are donated from part of the patients who participate this clinical research and may reflect the characteristics of the serum index. Readers can understand the difficulty of acquiring human bone samples, and they can understand even the trabecular microstructural analysis and mechanical properties are not fully agree with the serum characteristics. In this way, the serum characteristics are reliable based on large number, and then people will be willing to see some of the real bone microstructure change. Since you marked all the sample numbers clearly, this will not confuse the readers.

Author Response

Dear Review
We are pleased to receive the 3rd revision of our manuscript entitled “Microstructural and Strength Changes in Trabecular Bone in Elderly Patients with Type 2 Diabetes Mellitus” (ID -868209-R1)”, to be considered for publication. We have highlighted all changes in the text, clarifying comments in blue. In addition,

Reviewer #1 round 3:
Mercè Giner et al. have added a greater number of patients to analyze the serum parameters in the response letter. Thus, the Table 1 and Table 2 content has been substantially improved. The new version Table 1 and Table 2 are better for the paper since they reflect the real change with bigger sample number. I agree with the author that the importance and strength of this study is the precious human bone biopsies which allows us to "in situ" the bone microstructure of each selected group. In the paper, you can say the bone sample are donated from part of the patients who participate this clinical research and may reflect the characteristics of the serum index. Readers can understand the difficulty of acquiring human bone samples, and they can understand even the trabecular microstructural analysis and mechanical properties are not fully agree with the serum characteristics. In this way, the serum characteristics are reliable based on large number, and then people will be willing to see some of the real bone microstructure change. Since you marked all the sample numbers clearly, this will not confuse the readers.

A.R. Thank you very much for the comment. We agree with you, we think it is a good suggestion to add the new tables and add the comment “ you can say the bone sample are donated from part of the patients who participate this clinical research and may reflect the characteristics of the serum index ". For this, we have rectified the number of patients selected in the paper in:
- abstract (lines 20-23, page 1)
- M& M ( lines 74-78, page 2)
- Figure 1 (page 3)
. M&M (lines 131-133, page 4)

We have also modified the results, introducing the new values obtained with the new sample size in:
- line 168 (page 4)
- lines 170-176 (pages 4-5)
- lines 180-184 (page 5)
- Tables 1 and 2
- Discussion lines 286-289 (page 10)

This manuscript is a resubmission of an earlier submission. The following is a list of the peer review reports and author responses from that submission.

Round 1

Reviewer 1 Report

This cross-sectional study covers an interesting concept and is a novel area however, there are a few concerns.

Abstract does not do a good job of summarising as it provides insufficient summary of the methodology and no indication of results.

Consistency of certain terminology is missing such as bone mass density in line 40 and bone mineral density in line 45

Acknowledging the difficulty in recruitment, how can osteoarthritis patients be the reference group, used as healthy controls considering the physiological damage that those with osteoarthritis have on joints and overall function?

Please correct Dual-energy x-ray absorptiometry in methods

Please provide ethics approval number in methods

Author states an adjustment for age has been performed however, has not specified for which analysis please mention in statistical analysis section. Also, The mean age of osteoarthritis patients and osteoporosis patients was quite different how has this been accounted for?

This study includes unequal number of men and women how much would this have contributed to the findings, should be mentioned in limitations? 

Please fix BMI units in Table 1

Line 273 is unclear please rephrase

Reviewer 2 Report

Merce Giner et al, analyzed the trabecular bone microarchitecture and material properties using measurements of human bone biopsies from hip with and without fragility fracture in T2DM subjects. It is a hard study because it is difficult to obtain the samples, however, the design of experiment and the result have major flaws. I have some comments:

1, The sample number is too small for every group. It might be OK for a mouse study as litter mate, but for human study, with big variation and gender difference, the number is too small. It is shown that many of the SEM value in Table 1 are very big, which cause no significant difference in most of the comparison.

2, It is known that aging female have significantly lower bone mass than male, so it is not reasonable to mix the gender together to compare, especially when the sample number is small. Maybe the difference between OA and OP group is just because OP has higher percentage of female volunteer.

3, In Figure 2, compared with OA, the other 3 groups have lower BV/TV, trabecular thickness and trabecular number, but the trabecular separation is unchanged. How does the author explain it?

4, In Figure 3, the Young’s modulus data indicate that the bone samples in OA-T2DM, OP and OP-T2DM group are higher elastic than the OA group. How does the author explain it?

5, In Abstract, line 25, what does “importantly damaged” mean?

Reviewer 3 Report

This study was conducted to assess the impact of T2DM on trabecular bone BMD, microstructural and strength. I believe there are plenty of rooms for improvement on methodology of the study and how the results were interpreted. The following issues need to be addressed to improve the quality of manuscript.

Abstract: The statistical analysis results and p values should be presented. I don't believe the study conclusion is supported by the results. Authors stated that bone strength in the T2DM group was importantly (should be changed to significantly) damaged, confirming a worse quality and lower strength of the trabecular bone structure and mechanical properties. Study results supported this statement only in OA group (OAT2DM < OA, Fig 2 & 3). However, if taking all 4 study groups into consideration, OPT2DM is the least negatively impacted group, in reference to "OA without T2DM", in bone strength and microstructural changes (Fig 2 & 3). I was unable to see the significant differences between OPT2DM and OP groups in bone strength and microstructural parameters. In addition, the final statement in the abstract may not be supported by the study findings either. If the difference in bone strength and microstructure was only detected in the OA group, there were no fractures in OA group, how can readers be convinced that diabetes should be included in the FRAX model as a risk factor for osteoporotic fracture prediction?    

Methods: please provide details regarding how 23 patients were selected. Were 4 groups determined after 23 patients were selected or before? It looks like study groups were determined individually and patients were selected separately for each group, since the number of subjects between groups were pretty even, if that was the case, it should be a case-control study because OA as control group along with OP were determined first based on the clinical outcome (fractures) prior to analysis of BMD and bone strength. Cross-sectional study means 23 patients who undergone arthroplasty were selected before knowing the fracture or OA history. Please clarify the study design methodology. Convenience sampling should be listed as one of the study limitations.

In power calculation, please specify what effect measure “the standardized mean difference of 0.5” refers to (BMD, t-score, one of bone strength parameters?).

Sample size is very small with mixed gender in each group, whether women were on hormone therapy should be provided and adjusted in the statistical model.

Please provide version number of SPSS.

Table 1. Numerical difference was seen between OP and OP-T2DM in Hip/Femoral neck BMD and t-scores, although they are not statistically significant, it is likely a type 2 error. This could be one of the most critical flaws in this study which can bias the bone strength comparison results. In other words, if numerical difference in BMD between OP and OP-T2DM was truly statistically significant after increasing the sample size (type 2 error avoided), how can bone strength comparison between OP and OP-T2DM be valid to assess impact of T2DM on bone strength and microstructure?

If T2DM truly affects trabecular bone strength, there should be a detectable or at least numerical reduction in those bone strength parameters in OP-T2DM compared with OP group, however, study results indicated the opposite in Fig 2 & 3 which OP-T2DM seems to have a higher bone strength than OP group. This finding significantly contradicts the study conclusion that was solely drawn from the findings in OA population.

In line 251, authors reiterate the study findings by saying that “However, a lower quality of trabecular bone was not observed in OP-T2DM patients compared to non-diabetics.” Then in line 254, authors stated “These data note that diabetic disease is a key factor that is directly involved in the deterioration of bone quality”, which is not supported by the study findings in OP group, also contradicts the statement written in the line 251.

Conclusion: Once again, conclusion regarding any null findings in this study cannot be supported secondary to very small sample size (high risk of type 2 error), such as the statement in the conclusion “Furthermore, our results in OA-T2DM subjects confirm that diabetes does not impair BMD.”

The statement “Therefore, diabetes should be included as a risk factor of osteoporotic fracture in daily clinical practice and in the FRAX tool” can not be supported by the study findings unless the predictive value of T2DM in predicting fractures was assessed in this study.